

# Temperature effects on heavy rainfall modify catchment hydro-morphological response

Nadav Peleg[1], Chris Skinner[2], Simone Fatichi[1], Peter Molnar[1]

[1]Institute of Environmental Engineering, ETH Zurich, Zurich, Switzerland
[2]Energy and Environment Institute, University of Hull, UK

*Correspondence to*: Nadav Peleg (nadav.peleg@sccer-soe.ethz.ch)

**Abstract.** Heavy rainfall is expected to intensify with increasing temperatures, which will likely affect the rainfall spatial characteristics. The spatial variability of rainfall can affect streamflow and sediment transport volumes and peaks. Yet, the effect of climate change on the small-scale spatial structure of heavy rainfall and how those impacts hydrology and
geomorphology remains largely unexplored. In this study, the sensitivity of the hydro-morphological response to heavy rainfall at the small-scale of minutes and hundreds of meters was investigated. A numerical experiment was conducted, in which synthetic rainfall fields representing heavy rainfall events of two types, stratiform and convective, were simulated using a space-time rainfall generator model. The rainfall fields were modified to follow different spatial rainfall scenarios, associated with increasing temperatures, and used as inputs into a landscape evolution model. The experiment was conducted over a
complex topography medium-size (477 km$^2$) Alpine catchment in central Switzerland. The results highlight that the response of the streamflow and sediment yields are highly sensitive to changes in the rainfall structure at the small-scale, in particular to changes in the areal rainfall intensity and in the area of heavy rainfall, which alter the total rainfall volume, and to a lesser extent to changes in the peak rainfall intensity. The hydro-morphological response is enhanced (reduced) when the local peak rainfall intensified and the area of heavy rainfall increased (decreased). The hydro-morphological response was found to be
more sensitive to convective rainfall than stratiform rainfall because of localized runoff and erosion production. It is further shown that assuming heavy rainfall to intensify with increasing temperatures without introducing changes in the rainfall spatial structure might lead to over-estimation of future climate impacts on basin hydro-morphology.

## 1 Introduction

Changes in climate can impose modifications to fluvial systems that sometime exceed the historical natural variability (Blum
and Tornqvist, 2000; Fatichi et al., 2014; Goudie, 2006; Vandenberghe, 1995). These include, for example, changes to river streamflow, frequency and magnitude of floods, changes to the channel morphology and catchment connectivity and changes to the sediment yields (Bloschl et al., 2017; Coulthard et al., 2012b; Hancock, 2009; Lane et al., 2017; Tucker and Slingerland, 1997). The fluvial system is particularly sensitive to climate extremes, such as extreme rainfall events that can trigger landslides (Leonarduzzi et al., 2017), debris flows (Amponsah et al., 2016; Borga et al., 2014; Destro et al., 2018) or floods (Mallakpour
and Villarini, 2015; Marchi et al., 2010) that might rapidly change the landscape and stream morphology (Death et al., 2015; Krapesch et al., 2011; Thompson and Croke, 2013).



Spatio-temporal rainfall variability has been shown to play an important role in the hydro-morphological response of small to medium size catchments (i.e. in the order of $10^1$-$10^3$ km$^2$) affecting streamflow and sediment transport volumes, peaks and time to peaks (Arnaud et al., 2011; Bahat et al., 2009; Coulthard and Skinner, 2016; Kampf et al., 2016; Morin et al., 2006; Paschalis et al., 2014; Singh, 1997; Yakir and Morin, 2011; Zhu et al., 2018; Zoccatelli et al., 2011). Heavy rainfall events at

these scales have the potential to cover a given catchment entirely, thus increasing the sensitivity of the hydro-morphological response to the extreme event itself (Do et al., 2017; Sharma et al., 2018; Wasko and Sharma, 2017). The impact of rainfall variability on the sensitivity of the hydro-morphological response is more significant in climate regimes where a substantial part of the rainfall is associated with convective events (Belachsen et al., 2017; Goodrich et al., 1995; Kampf et al., 2016; Peleg and Morin, 2012; Wright et al., 2013), and is most pronounced when heavy rainfall events are considered (Marra and

Morin, 2018; Peleg et al., 2018b). Therefore, rainfall fields at high spatial and temporal resolution, at the scales appropriate to simulate rainfall convective features (i.e. 1 km and 10 min, or finer), are needed for hydrological and geomorphological climate change impact studies (Coulthard and Skinner, 2016; Gires et al., 2015; Li and Fang, 2016; Morin et al., 2006; Ochoa-Rodriguez et al., 2015; Peleg et al., 2015; Skinner et al., 2019; Zhu et al., 2018).

Changes in heavy rainfall in recent decades, such as extremely long wet spells and rainfall intensification, have been reported

in different regions (Alexander et al., 2006; Fischer and Knutti, 2016; Peterson et al., 2013; Singh et al., 2014; Westra et al., 2013). The intensity of heavy rainstorms is sensitive to warming (e.g. Berg et al., 2013; Molnar et al., 2015) due to warmer air having an increased water vapor holding capacity, which in saturated conditions follows the Clausius-Clapeyron (CC) relationship (O'Gorman and Schneider, 2009; Trenberth et al., 2003). The characteristics of heavy rainfall, such as intensity, frequency and duration, are foreseen to continue changing as a consequence of increasing emissions of anthropogenic

greenhouse gases and thus increasing temperatures in the future (Fischer et al., 2013; Fischer and Knutti, 2015; Orlowsky and Seneviratne, 2012).

Only few studies have used observed data, obtained from dense rain-gauge networks or from weather radar estimates, to analyse the impact of increasing temperatures on the spatial characteristics of heavy rainfall (e.g. Berg et al., 2013; Lochbihler et al., 2017; Peleg et al., 2018a; Wasko et al., 2016). Although similar methods were employed, different results were reported

for various regions and climates. Wasko et al. (2016) reported a re-distribution of available humidity from the low intensity regions of the rain field toward the high-intensity regions for tropical, temperate and arid climates in Australia, meaning that while the peak rainfall of the storm is intensifying, the area of the heavy rainfall reduces with increasing temperatures (case 1, Fig. 1). Peleg et al. (2018a) observed similar trends of changes in spatial rainfall characteristics for Mediterranean climate, but for semi-arid to arid climates (east Mediterranean region) they found that while higher temperatures lead to an increase in the

peak intensity of heavy rainfall, the area of the heavy rainfall remains largely unchanged or slightly reduces, with a small weakening in total rainfall amounts (case 2, Fig. 1). Lochbihler et al. (2017) observed that both the area and the intensity of heavy rainfall increase with rising temperatures for temperate maritime climate in the Netherlands (case 3, Fig. 1). Apparently, the effect of temperature on the small-scale spatial structure of heavy rainfall varies across locations, likely due to differences





in climate dynamic conditions and available humidity (Pfahl et al., 2017), and remains largely unexplored for many regions worldwide.

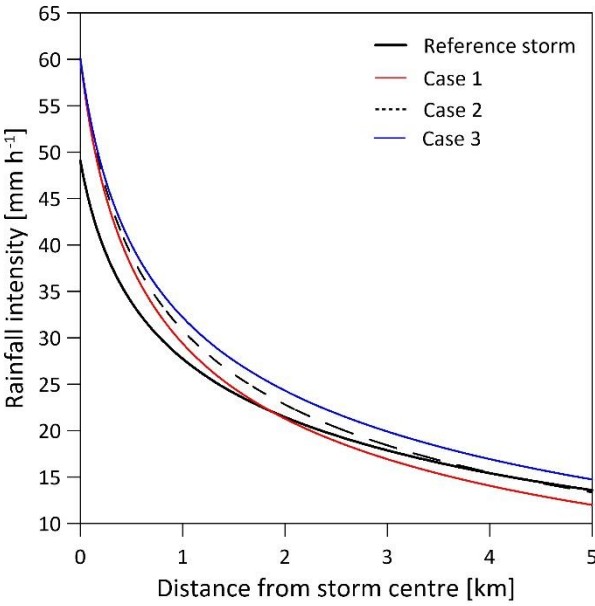

**Figure 1. Schematic illustration of the changes to the spatial structure of storms with increasing temperature. The peak of the storm is at the storm's center and the rainfall intensities follow a log-normal decay with distance from the center. The black line represents a storm for relatively low temperature (reference storm). Other lines (cases 1 to 3) represent plausible storms at higher temperature with intensified rainfall peak, but different areal rainfall and area of heavy rainfall (see text for details).**

The future space-time structure of heavy rainfall at the small-scale can be inferred from two main sources. The first option is by simulating rainfall using a Convection-Permitting Model (CPM). Convective processes are represented explicitly in CPMs based on the governing dynamical equations (Ban et al., 2014, 2015; Prein et al., 2015) that allow the representation of the space-time structure of rainfall directly at small-scale, i.e. without the need of further downscaling or applying de-biasing methods. Prein et al. (2017) for example, used a CPM to simulate how the rainfall space-time structure is changing in future

climate over the US. However, the CPM approach for simulating future rainfall comes with a drawback: it is highly demanding in terms of computational resources because high-performance computing is needed to run the models. The second option is to use space-time rainfall generator models (Benoit et al., 2018a; Paschalis et al., 2013; Peleg et al., 2017b; Peleg and Morin, 2014; Singer et al., 2018) to simulate rainfall based on information derived from weather radar for the present and Regional Climate Models (RCM) for the future. RCMs simulate rainfall fields at a spatial resolution not far from what is needed in local

impact studies (e.g. some Euro-Cordex models are at 11-km resolution, Jacob et al., 2014), but they do not resolve convection processes explicitly. Changes in rainfall simulated by RCMs should be combined with proper observations of the space-time rainfall structure at small-scale, obtained from a dense rain-gauge network or a weather radar (Benoit et al., 2018b; O and

Foelsche, 2018; Peleg et al., 2013), and from the relationship between the rainfall spatial properties and other climate variables like near-surface air temperature or dew point temperature (Berg et al., 2013; Mishra et al., 2012; Molnar et al., 2015; Westra et al., 2014). The main shortcomings of this second alternative is that the information needed is not readily available for many locations and that the approach relies on the strong assumption that the rainfall-temperature relationships of the present climate

holds true for the future (Peleg et al., 2019).

Therefore to investigate the impacts of temperature-induced changes in rainfall on hydro-morphological response of catchments expert knowledge in setting and operating climate models or rainfall generator models is required. This is likely the reason why the question of the sensitivity of hydro-morphological response to spatial changes in high-resolution rainfall fields has not been extensively explored so far, and only few studies have analysed geomorphological implications of climate

impacts using stochastic approaches or distributed rainfall (e.g. Coulthard and Skinner, 2016; Francipane et al., 2015). The 'geomorphic multiplier' concept (Coulthard et al., 2012b), i.e. the non-linear relation between streamflow and sediment yield, emphasizes the importance of answering the question at hand, as the effects of the rainfall structure on the hydrology are relatively studied, yet the effects on sediment production and transport are still largely unknown.

In this study, we aim at exploring the sensitivity of the hydro-morphological response to rainfall at the convective scale of

minutes and hundreds of meters, which are the relevant scales for the hydrological response of small and medium-size catchments. To this end, a numerical experiment was conducted, in which synthetic rainfall fields representing a characteristic heavy rainfall event were simulated. The rainfall fields were then modified to follow different spatial rainfall scenarios, associated with increasing temperatures. The numerical experiment was conducted over a medium size Alpine catchment with a complex topography (Kleine Emme, central Switzerland, 477 km$^2$). The sensitivity of the hydro-morphological response to

the rainfall spatial properties, the implications for climate change impact studies, the generalization of the results, and the limitations of the numerical experiment are discussed. The approach also provides an example of how uncertainties in climate change impact assessments and their propagation into hydro-morphological response should be properly framed.

## 2 Methods

### 2.1 Experimental setup

The sensitivity of catchment hydro-morphological response to changes in the spatial properties of rainfall intensity during heavy rainfall events was examined using a combination of models. The numerical experiment is composed of two steps. First, a time series of mean areal rainfall over the domain is simulated. The time series follows a temporal structure of a design storm characterized by a Gaussian shape that is assumed representative of a heavy rainfall event for present climate conditions (Fig. 2). A rainfall generator model (Section 2.2) is then used to generate multiple realizations of gridded rainfall for the design

storm. In practice, the rainfall generator downscales the areal rainfall over the domain to a finer spatial resolution. Each of the realizations has a different 2-dimensional representation of the rainfall intensities in space. In this way, the spatial rainfall variability (stochasticity) is explicitly accounted for (see further discussion on the role of stochastic rainfall spatial variability





by Paschalis et al., 2014; Peleg et al., 2017a, 2018b, 2019). The model was set to simulate two types of rainfall, stratiform and convective.

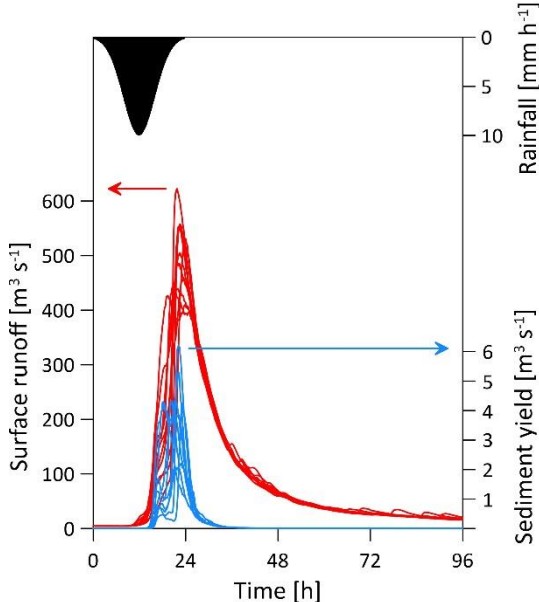

**Figure 2. Illustration of the rainfall intensity over the entire area of the design storm (black area) and the respectively generated stream runoff (red lines) and sediment yields (blue lines) from 10 stochastic rainfall realizations produced with the rainfall generator.**

The rainfall fields are then modified using the rainfall generator to follow 4 "temperature increase scenarios" and 4 "areal rainfall scenarios" (16 spatial scenarios altogether, see Table 1). In all of the scenarios, the peak rainfall intensity at the grid scale (i.e. the grid cells with the maximum rainfall intensities at the time the areal rainfall peaks) is assumed to intensify at a rate of 7% °C$^{-1}$, which corresponds with a fully saturated air column and is the CC rate (Trenberth et al., 2003). The areal rainfall scenarios refer to 4 cases: (i) a decrease in the mean areal rainfall (by -3% °C$^{-1}$) that is followed by a decrease in the area of heavy rainfall; (ii) no change in the mean areal rainfall (0% °C$^{-1}$), but a small decrease in the area of heavy rainfall; (iii) an increase in the mean areal rainfall (by 3% °C$^{-1}$) with a small increase in the area of heavy rainfall; and (iv) an increase in the mean areal rainfall of the same rate as peak rainfall (7% °C$^{-1}$), with a significant increase of the area of heavy rainfall (Table 1 and Fig. 1).

Peak rainfall intensity can intensify at a higher rate than 7% °C$^{-1}$, potentially increasing the consequences on the fluvial response. To further demonstrate this point, part of the numerical experiment (case 2, convective rainfall type) was extended to include intensification of the peak rainfall at the grid scale from the CC rate (7% °C$^{-1}$) to the super-CC rate (14% °C$^{-1}$, e.g. Lenderink and Van Meijgaard (2008), 2CC from hereafter).



**Table 1. The 4 "temperature increase scenarios" (ΔT), and the 4 scenarios for changes in the mean areal rainfall (expressed by % °C⁻¹). Colors represent qualitatively changes in the area of the intense rainfall, from a large increase (dark blue) to a large decrease (dark green). For all the scenarios the peak rainfall intensity is assumed to increase at a rate of 7% °C⁻¹.**

| $\Delta T=1°C$ | $\Delta T=2°C$ | $\Delta T=3°C$ | $\Delta T=4°C$ |
|---|---|---|---|
| 7% °C⁻¹ | 7% °C⁻¹ | 7% °C⁻¹ | 7% °C⁻¹ |
| 3% °C⁻¹ | 3% °C⁻¹ | 3% °C⁻¹ | 3% °C⁻¹ |
| 0% °C⁻¹ | 0% °C⁻¹ | 0% °C⁻¹ | 0% °C⁻¹ |
| -3% °C⁻¹ | -3% °C⁻¹ | -3% °C⁻¹ | -3% °C⁻¹ |

5    An example of simulated rainfall fields for the 4 spatial rainfall scenarios, for the case with a temperature increase of 3°C, is presented in Fig. 3. The effects of the changes in the spatial structure of the rainfall on the spatial correlation of the field are shown in Fig. 4.

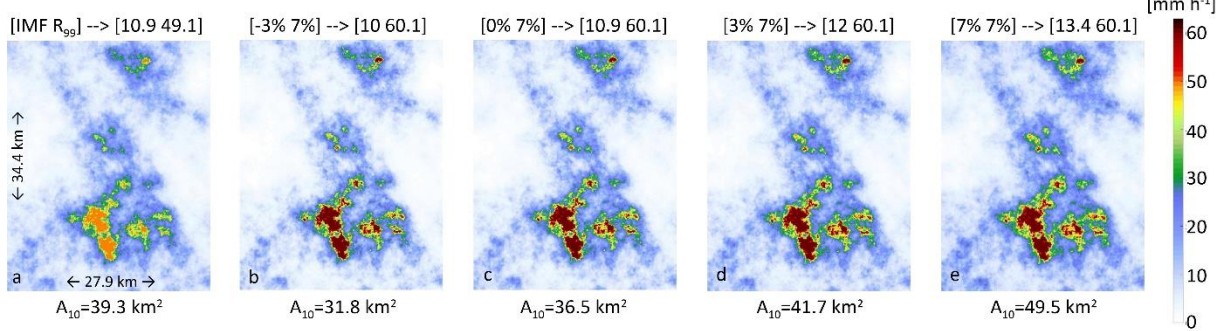

10   **Figure 3. (a) An example of a simulated rainfall field for a given time step. (b-e) plots of the rainfall field with 4 different "areal rainfall scenarios" (Table 1) and for a specific "temperature scenario" of ΔT=3°C. IMF refers to changes in the mean areal rainfall [% °C⁻¹] and R$_{99}$ refer to changes in the peak rainfall intensity [% °C⁻¹] in comparison to (a) and the absolute values of the change are given in parentheses [mm h⁻¹]. A$_{10}$ refers to the total area [km²] above a rain intensity threshold of 10 mm h⁻¹.**




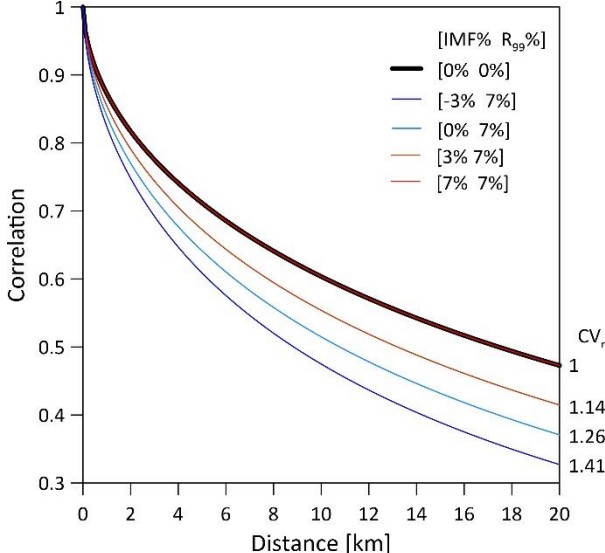

**Figure 4. Spatial correlation of the rainfall fields that are presented in Fig. 3 for the specific case of ΔT=3°C. The correlation is computed on the spectral densities of the fields following (Peleg et al., 2019) and for each case the spatial coefficient of variation of the field $CV_r$ is given. IMF refers to changes in the mean areal rainfall [% °C⁻¹] and $R_{99}$ refers to changes in the peak rainfall intensity [% °C⁻¹].**

In the second stage, the multiple stochastic realizations of design storms that were simulated for each of the 16 different scenarios and for the 2CC experiment were fed into a landscape evolution model (Section 2.3) to simulate the hydro-morphological response, i.e. streamflow and sediment transport (Fig. 2). The sensitivity of the response to the changes in rainfall spatial properties was finally analyzed.

## 2.2 Space-time stochastic rainfall generator model

Gridded stochastic rainfall generator models can be used to generate multiple realizations of a given (design) storm (McRobie et al., 2013; Paschalis et al., 2014; Peleg et al., 2018b; Shah et al., 1996a, b). The realizations preserve the temporal evolution of the mean areal rainfall over the domain, but differ in how the rainfall intensities are distributed in space within the domain. Here a simplified version of the STREAP (Space-Time Realizations of Areal Precipitation) rainfall generator model was used for generating high-resolution space-time rainfall fields (Paschalis et al., 2013). STREAP is a substantial improvement of previous space-time rainfall generators (Bell, 1987; Kundu and Bell, 2003; Pegram and Clothier, 2001b, a) and was recently further developed by Peleg et al. (2017b). The model simulates rainfall fields in three steps: (i) the length of the storms and the intra-storm periods are generated; (ii) the temporal evolution of the mean areal intensity over the domain and the fraction of wet areas are simulated; and (iii) these time series are translated into intermittent rainfall fields. As the design storm used in this study is predefined, only step (iii) is required in this study. Several modifications to this step were required in order to tailor the spatial structure of the rainfall fields to follow prescribed changes in both the peak and areal rainfall with temperature.





The STREAP model is discussed in detail by Paschalis et al. (2013) and here a brief description of step (iii) and specific modifications used in this case study are presented.

The intermittent rainfall field is simulated as a probability transformation of an isotropic Gaussian random field that is computed using the Fast Fourier Transform method. As in previous studies, we assume that rainfall intensity is spatially

5  distributed following a lognormal distribution (e.g. Paschalis et al., 2013, 2014; Peleg et al., 2017b, 2018b, 2019). A lognormal function is therefore applied to convert the isotropic Gaussian field to the intermittent rainfall field. The information needed for this transformation is the mean areal rainfall over the domain (abbreviated hereafter as $IMF$, following the notations of Pegram and Clothier, 2001a) and the rainfall coefficient of variation ($CV_r$) in space, which is a model parameter. Assuming that the rainfall covers the entire domain (i.e. the wet area ratio $WAR$ is equal to 1, see Paschalis et al. (2013) for details), the

10  intermittent rainfall fields are expressed as:

$$R(x, y, t) = LN^{-1}(U[G(x, y, t)], \mu_r, \sigma_r), \quad (1)$$

where $R(x, y, t)$ is the rainfall intensity in space and time, $LN^{-1}$ is the inverse cumulative lognormal distribution and $U[G(x, y, t)]$ are the percentiles in space and time of the latent isotropic Gaussian field. The parameters of the lognormal distribution, $\mu_r$ and $\sigma_r$ are expressed as:

$$\mu_r = log\left(\frac{IMF(t)}{\sqrt{CV_r^2 + 1}}\right), \quad (2)$$

and

$$\sigma_r = \sqrt{log(CV_r^2 + 1)} \quad (3)$$

The peak rainfall intensity at the grid scale is defined as the inverse lognormal of the 99[th] percentile, notated as $R_{99}$ and expressed as:

$$R_{99}(t) = e^{log\left(\frac{IMF(t)}{\sqrt{CV_r^2 + 1}}\right) + \sqrt{log(CV_r^2 + 1)}\left[\sqrt{2}erf^{-1}(2 \times 0.99 - 1)\right]} \quad (4)$$

Eq. (4) can be simplified to find a unique relation between the mean areal rainfall intensity and the peak rainfall intensity at each time step:

$$log\left(\frac{R_{99}(t)}{IMF(t)}\right) = 2.3263\sqrt{log(CV_r^2 + 1)} - log\left(\sqrt{CV_r^2 + 1}\right) \quad (5)$$

The rainfall spatial coefficient of variation is a model parameter that changes depending on the scenarios. If, for example, the

25  $CV_r$ value for the reference simulation (i.e. ΔT=0°C) is equal to 1, the $CV_r$ values will increase as the difference between the scaling of the peak rainfall intensity and the areal rainfall increases. An example of how $CV_r$ changes as a function of the scaling of the peak rainfall and the areal rainfall intensities is presented in Fig. 4.





## 2.3 Hydro-morphological model

The hydrological and geomorphic response to rainfall is explored using a Landscape Evolution Model (LEM, see review paper by Tucker and Hancock, 2010). The CAESAR (Coulthard et al., 2002) and CAESAR-Lisflood models (Coulthard et al., 2013) are grid-based LEMs that have been widely used to simulate morphodynamic changes over short temporal (<1 y or event based) and small spatial (<1 km$^2$) scales (Coulthard et al., 2012a; Hoober et al., 2017) to long temporal (>10$^4$ y) and large spatial (10$^3$ km$^2$) scales (Coulthard and Van De Wiel, 2017; Hancock et al., 2010, 2015). Both models were used in the past to explore hydro-morphologic sensitivity to climate change and rainfall temporal variability (Coulthard et al., 2012b; Hancock, 2009, 2012; Hancock and Coulthard, 2012; Hoober et al., 2017). Recent versions of the CAESAR-Lisflood model have the ability to use gridded rainfall as an input (Coulthard and Skinner, 2016; Skinner et al., 2019), which is an essential ability for this study. This feature is lacking in many other LEMs. Surface runoff in the model is computed using the TOPMODEL hydrological model (Beven and Kirkby, 1979) and is routed downstream using the LISFLOOD-FP model (Bates et al., 2010), which generates flow depths and velocities. Fluvial erosion is simulated using either Wilcock and Crowe, Einstein, or Meyer-Peter-Muller equations, moving sediments that are stored in an active-layer system that can handle up to 9 grain sizes. Lateral erosion, slope processes, soil development and interaction with vegetation are also simulated by the model. Version "1.9h" of the model was used in this study, and no modifications were made to the source code. For further details on the model, the reader is referred to Coulthard et al. (2013).

## 3 Study catchment

A medium size catchment (i.e. in the order of 10$^2$-10$^3$ km$^2$) is the most suitable case study for the designed numerical experiment. This can be either a synthetic or a real catchment. There are advantages of using a synthetic domain where the effects of different catchment properties (e.g. area, orientation, aspect) can be separated and individually examined (e.g. Mastrotheodoros et al., 2019). However, the main benefit of using a real catchment is that the LEM outputs can be validated against observations, in order to evaluate the model suitability in simulating the hydro-geomorphological response. In this study, we calibrated the model in a real catchment with observed data, but simplified the catchment and reduced the complexity of the simulation in order to eliminate possible effects of other variables than rainfall (such as vegetation-erosion interactions) on the sensitivity of the hydro-morphological response. The model parameterization is discussed in Section 3.2.

The study was conducted over the Kleine Emme catchment (Fig. 5), located in central Switzerland (8°E 47°N). There are several reasons for the selection of this catchment: (i) intense convective rainfall events are common over the region during summer and rainfall is associated with high space-time variability (Isotta et al., 2014; Molnar et al., 2015; Panziera et al., 2018); (ii) the catchment is well monitored in terms of rainfall and streamflow, including the extreme rainfall and flood event that occurred in August 2005 with an estimated return period exceeding 100 y (Beniston, 2006; Jaeggi, 2008; Rickenmann et al., 2016; Rickenmann and Koschni, 2010); (iii) the hydrology and geomorphology were successfully explored using numerical models for this catchment in the past (e.g. Heimann et al., 2015; Paschalis et al., 2014); (iv) the streamflow is close to natural



conditions (i.e. without irrigation or hydropower uses) and the catchment is glacier-free; and (v) the catchment is representative in terms of topographic (area of 477 km², elevation range between 430 and 2300 m above sea level), hydrological and geomorphological features of a typical Alpine catchment.

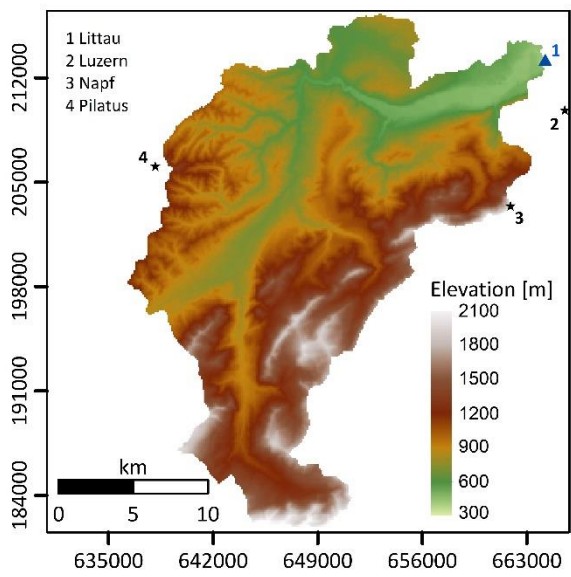

**Figure 5. Topographic map of the Kleine Emme catchment. The blue triangle marks the location of the hydrometric station and the black stars mark the locations of the meteorological stations that were used in this study. Coordinates are in the Swiss reference system CH1903 [km].**

**3.1 Data**

We set the numerical models to simulate the impacts of a heavy rainfall event that is similar to the event that occurred in August 2005 (data were collected for the period between 14[th] and 28[th] of August). Rainfall data were gathered from the Swiss Federal Office for Meteorology and Climatology (MeteoSwiss) from two different products. Rainfall records at 10 min temporal resolution were obtained from three MeteoSwiss - SwissMetNet ground-stations surrounding the catchment (Fig. 5). Mean areal rainfall over the domain was computed by averaging rainfall from the three locations. It was then temporally downscaled to 5 min resolution using a simple linear interpolation. We assume that by using the data from these three stations the temporal dynamics of the mean areal rainfall over the catchment is adequately represented (i.e. the timing of the storm is preserved). However, the mean areal rainfall intensities might not be representative for the entire catchment as the three meteorological stations are located in the northern part of the catchment (Fig. 5). Therefore, the rainfall amounts were corrected using data from gridded 1 km daily rainfall estimates that were derived from MeteoSwiss product RhiresD (MeteoSwiss, 2016; Schwarb, 2000) that covers the entire catchment. For each day, the mean areal rainfall over the catchment was computed from





the RhiresD product. The 5 min rainfall intensities were aggregated to the daily scale, compared with the daily RhiresD estimates and corrected using a multiplicative factor based on the ratio between the two. The final time series of the mean areal rainfall is presented in Fig. 6.

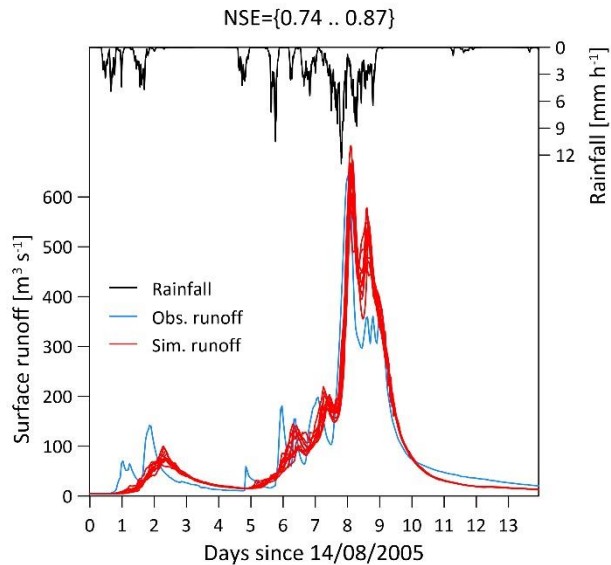

**Figure 6. Illustration of the heavy rainfall event of August 2005 in the Kleine Emme catchment. The black line represents the mean areal rainfall over the catchment. The blue and red lines represent the observed and simulated hydrographs, respectively, at the outlet of the catchment. The 10 simulated lines are a result of the stochastic downscaling of the areal rainfall to a finer (100 m) spatial resolution using the rainfall generator model and its range represents the sensitivity of the hydrological response to the spatial**
**rainfall variability.**

Hourly streamflow data were supplied by the Swiss Federal Office for the Environment for the Emmen station located at the outlet of the catchment (Fig. 5). Estimates of the sediment yield, grain size distribution, and the geomorphologic response to heavy rainfall events for the Kleine Emme, and for nearby catchments that are representative of the study area, were obtained
from multiple sources (Bezzola and Hegg, 2008; Heimann et al., 2015; Rickenmann et al., 2016, 2008; Rickenmann and Koschni, 2010; Rickenmann and McArdell, 2007; Steeb et al., 2017). The digital elevation map (Fig. 5) and bedrock map (Fig. S1) were supplied by the Swiss Federal Office of Topography at 25 m resolution and were upscaled to 100 m resolution, which was the resolution used for model simulations. The surface roughness map (Fig. S2) at 100 m resolution was prepared based on land use map that was obtained from the Swiss Federal Statistical Office following the classification suggested by (Te
Chow, 1959).



## 3.2 Setup of the rainfall generator model

The parameters of the rainfall generator model (Table 2) were not calibrated to reproduce the statistics of a specific heavy rainfall event that was observed over the catchment, but to simulate realistic space-time structure of rainfall fields over the Alpine region. The rainfall fields are simulated to move with a constant speed and direction as the hydrological response is

sensitive to these parameters (Paschalis et al., 2014; Yakir and Morin, 2011). The parameters for the correlation length, storm velocity and direction follows reference values that were found by analyzing rainfall fields obtained from weather radar system in nearby Alpine catchments (Peleg et al., 2017a, 2017b, 2019). The $CV_r$ parameter was set to be equal to 1 for the simulations of the real storm and for the simulation of the basic design storm (for $\Delta T=0$), and was changed depending on the different scaling scenarios (see Section 2.2 and Fig. 4). The wet area ratio is assumed equal to 1 for all simulations, as almost the entire

catchment is covered with rainfall during heavy rainfall events (Paschalis et al., 2014).

**Table 2. Parameters of the stochastic rainfall generator model.**

| Parameter | Units | Value |
|---|---|---|
| Wet area ratio | - | 1 |
| Correlation length | km | 30 |
| Storm velocity | km h⁻¹ | 15 |
| Storm direction | ° from north | 90 |
| Coefficient of variation of spatial rainfall | - | 1 |

The rainfall generator model is used to downscale the rainfall from its averaged value over the domain (i.e. mean areal rainfall)

to a rainfall field containing 344 x 279 grid cells at a fine resolution of 100 m in space (see example in Fig. 3). Over the domain, the generated rainfall fields preserve the value of the mean areal rainfall at each time step exactly. As the catchment is smaller than the domain extent, the areal rainfall computed over the catchment is likely to be a bit smaller too (but note that the spatial rainfall scenarios refer to the changes in the areal rainfall for the domain extent). For each of the storms that were downscaled (the "real" storm of August 2005 and the different scenarios of the design storm), an ensemble of 10 realizations was

stochastically generated in order to account for the natural spatial variability of the rainfall.

A design storm capable of triggering a significant streamflow and mass movement over the catchment is required for the numerical experiment. Therefore, the storm was designed (i) for a long duration of 24 h, corresponding to the 90[th] percentile of storms' duration for this catchment (Paschalis et al., 2014); and (ii) to reach a mean areal rainfall maximum of 10 mm h⁻¹, which is slightly lower than the mean areal rainfall maximum that is reported for the August 2005 heavy rainfall event [between

13 mm h⁻¹ as computed here and 14 mm h⁻¹ as reported by Rickenmann and Koschni (2010) and by Steeb et al. (2017)], but still higher than the estimated value for a 10 y return period (Paschalis et al., 2014). Two types of rainfall were examined, stratiform and convective, that have different spatial characteristics but are both typical for an Alpine region (Gaal et al., 2014). In general, stratiform rainfall is characterized with lower rainfall peaks and is more homogenous in space in comparison to



convective rainfall (Benoit et al., 2018b; Panziera et al., 2015, 2018). For the initial setting (ΔT=0) of the stratiform rainfall type, the rainfall peak at the grid scale (100 m and 5 min) was set for 49 mm h$^{-1}$ - a rainfall peak that is estimated to be around the 2 y return period when comparing to MeteoSwiss stations in the region (e.g. the peak rainfall in Pilatus station for 2 y return period and 10 min duration is 54 mm h$^{-1}$). In addition, a $CV_r$ value of 1 (rather homogenous rainfall field) was set. For

the initial setting of the convective rainfall type, the rainfall peak at the grid scale was set to 120 mm h$^{-1}$ (corresponds to 30 y return period) and a $CV_r$ value of 3.85 corresponding to a non-homogenous field, typical of "convective cell formation" was set.

### 3.3 Setup of the LEM

To reduce the complexity of the CAESAR-Lisflood model, several features were not considered, such as lateral erosion (Van

De Wiel et al., 2007) and vegetation effects on erosion (Hancock et al., 2015). The parameters of the CAESAR-Lisflood model (Table 3) were calibrated based on the August 2005 heavy rainfall event (Fig. 6). The model was used in catchment (basin) mode (Coulthard et al., 2013) using the Einstein (Einstein, 1950) formulation to compute sediment transport. The gridded rainfall realizations that were simulated by the rainfall generator were used as inputs into the model (as in Coulthard and Skinner, 2016; Skinner et al., 2019) and the model was set for the same spatial (100 m) and temporal (maximum dynamic time

step of 5 min) resolution of the rainfall fields. The model parameters are the same for all grid cells in the domain, except for the Manning coefficient values, which are spatially distributed (Fig. S2). Evaporation rate (Fatichi et al., 2015), and the values of grain size distribution and proportion (Rickenmann and McArdell, 2007), follow estimates and observations from nearby Alpine areas.

**Table 3. Parameters of the CAESAR-Lisflood model.**

| Parameter | Units | Values |
| --- | --- | --- |
| Grainsizes | m | 0.00035, 0.003, 0.016, 0.04, 0.125, 0.2 |
| Grainsize proportions (sum to 1) | - | 0.1, 0.2, 0.2, 0.2, 0.2, 0.1 |
| Suspended sediment fall velocity | m s$^{-1}$ | 0.045 |
| Sediment transport law | - | Einstein |
| Max erode limit | m | 0.005 |
| Active layer thickness | m | 0.1 |
| TOPMODEL 'm' value | - | 0.01 |
| Water depth threshold for erosion | m | 0.01 |
| Courant number | - | 0.3 |
| Mannings coefficient | - | 0.01-0.2 (distributed, see Fig. S2) |

The hydrological TOPMODEL parameter 'm' (Beven and Kirkby, 1979) and the Courant number (Bates et al., 2010) were calibrated by finding the optimal fit between the simulated 14 day-long hydrographs and the observed hydrograph (Fig. 6). The model was first run for a "spin up" period of one year to eliminate sharp gradient changes in the elevation and to





redistribute the grainsize distributions along the channels from the initial global setup described in Table 3. The calibration resulted in model efficiency (Nash and Sutcliffe, 1970) ranging between 0.74 and 0.87 as a result of different stochastic rainfall fields. The model also simulates well the total streamflow volume and peak streamflow, as both observed points fall within the 25th-75th percentile range of the 10 simulated realizations (Fig. 7).

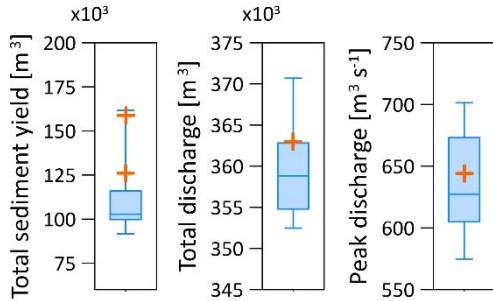

**Figure 7. Box-plots showing the median (horizontal line), 25th-75th percentile range (shaded area) and maximum and minimum range (bounded with lines) of the total sediment yield (left), total streamflow volume (centre) and peak streamflow (right) from the 10 downscaled spatial rainfall realizations for the heavy rainfall event of August 2005. The orange '+' symbols represent the observed**
**data for the total and peak streamflows, and the estimated range of the total sediment yield.**

The impacts of the changes in the spatial structure of heavy rainfall are expressed on bedload and suspended sediment load. For the August 2005 event, the volume of the sediment transport at the outlet of the catchment (minus the observed volume estimated for the lateral erosion that is not simulated by the model) was estimated between 125,000 m$^3$ (Bezzola and Hegg, 2008) and 160,000 m$^3$ (Rickenmann et al., 2016; Rickenmann and Koschni, 2010). Most simulated realizations underestimate
the reported volumes (25th-75th percentile range, Fig. 7), yet two of the realizations do fall within this range (simulating 128,620 and 161,600 m$^3$ of sediment yield).

Each individual simulation of the design storm experiment runs for an 8-days period. The first 4 days are used as a spin-up period to stabilise the streamflow at the outlet of the catchment for a baseflow of 10 m$^3$ s$^{-1}$. Therefore, model output analysis for the runs of the design storm starts after day 4 of the simulation.

**4 Results**

The sensitivity of the hydro-morphological response to the changes in rainfall spatial structure was examined in terms of peak streamflow and sediment yield, for the total volumes of streamflow and sediment yield and for the inundated area and the area subjected to erosion or deposition. For each of these components, the mean value from the 10 simulated realizations was computed for each of the scenarios. Results are presented first for stratiform rainfall, then for convective rainfall, followed by
a summary of the overall findings.





## 4.1 Stratiform rainfall type

Peak streamflow and sediment yield - as expected - are predicted to enhance (reduce) with increasing (decreasing) rainfall amounts, however the peaks were found to be mainly sensitive to the changes in the mean areal rainfall, since changing rainfall peak intensity only did not affect streamflow considerably. For example, examining the results for $\Delta T = 4°C$ in Fig. 8, an

increase of the mean areal rainfall by 7% $°C^{-1}$ resulted in an increase in peak streamflow by 62%, an increase of the mean areal rainfall at a rate of 3% $°C^{-1}$ resulted in an increase in peak streamflow by 25%, and a decrease in the mean areal rainfall at a rate of -3% $°C^{-1}$ resulted in a decrease in peak streamflow by -18%. The peak sediment yield also showed higher sensitivity to changes in mean areal rainfall (Fig. 8, changes to the symbol sizes). For example, for $\Delta T = 4°C$, an increase of the mean areal rainfall by 7% $°C^{-1}$ and 3% $°C^{-1}$ resulted in an increase of 210% and 84% (respectively) in the peak sediment yield, and a

decrease in the mean areal rainfall at a rate of -3% $°C^{-1}$ resulted in a decrease in sediment yield peak by -31%, underlying how non-linearities in sediment transport are much stronger than for discharge.

Focusing on the results where the peak rainfall intensity at the grid scale is intensifying by 7% $°C^{-1}$ but the mean areal rainfall remains unchanged (Fig. 8, orange symbols) reveal different sensitivities for the peak streamflow and peak sediment yield. Although the peak rainfall at the grid scale intensified, peak streamflow is hardly affected; the maximum enhancement is 2%

for the $\Delta T = 4°C$ scenario, which is almost negligible. However, the intensification of the peak rainfall at the grid scale resulted in a considerable enhancement of the peak sediment yield of 4% for $\Delta T = 1°C$, 8% for $\Delta T = 2°C$, 11.8% for $\Delta T = 3°C$ and 16.4% for $\Delta T = 4°C$ (Fig. 8).

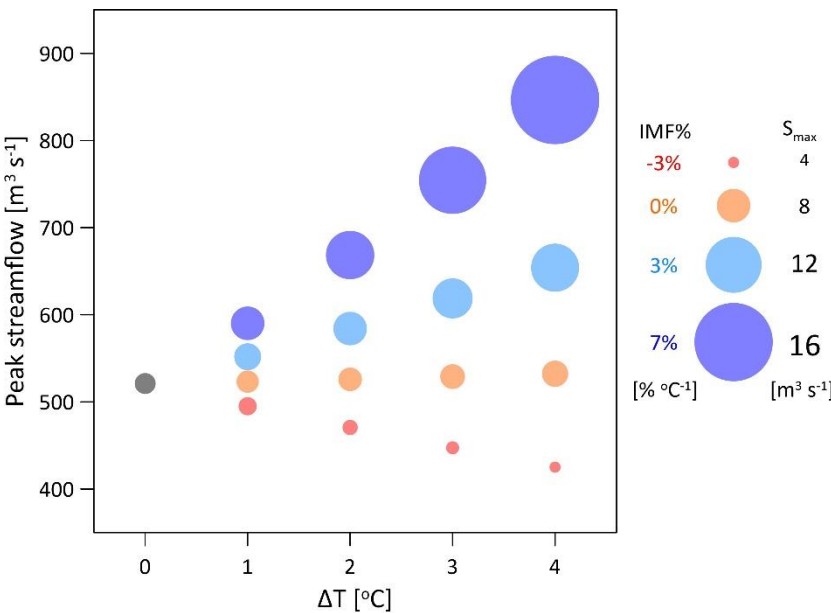

**Figure 8. Response of the peak streamflow (y-axis) and peak sediment yield (symbol size, $S_{max}$) at the outlet of the catchment to changes in the rainfall spatial structure with temperature. Peak rainfall intensity increases by 7% $°C^{-1}$ for all points. Different colors**





**represent a mean areal rainfall intensity decrease of -3% °C⁻¹ (red), not changing (orange), and increase of 3% °C⁻¹ (blue) and 7% °C⁻¹ (purple). Reference (base scenario) for the sensitivity is the single point at ΔT=0 (black).**

Similar sensitivities were found when examining the effects of the intensification of the peak rainfall intensity at the grid scale

on total streamflow and sediment yield, assuming no changes to the mean areal rainfall (case 2). While total streamflow changed by less than 1% for the different temperature scenarios, total sediment yields increased by 2%, 4%, 7% and 9% for the ΔT=1°C to 4°C scenarios, respectively. However, modifying the mean areal rainfall, even by simply applying ±3% °C⁻¹, resulted in a change to the total sediment yields higher than 10% (Fig. 9).

The 'geomorphic multiplier' implies a non-linear relation between total streamflow and total sediment yield (Coulthard et al.,

2012b). Plotting these against each other (Fig. 9), using information from all the scenarios, the sensitivity of the 'geomorphic multiplier' to changes in the rainfall spatial structure can be explored. The relation between the total sediment yield and the total streamflow were found to follow a power law once fitted to the actual simulated values of the total streamflow and sediment yield ($R^2$=0.99, not shown), which is positively correlated to the changes in total rainfall amounts (ΔP, symbol size). Most points fall directly along the fitted line, which implies that the total streamflow and total sediment yield are mainly

sensitive to changes in the total rainfall volume. Points that deviate from the fitted line, e.g. the scatter of dots along the y-axis close to the 0% change in total streamflow, have different spatial structures of rainfall. The sensitivity of the total sediment yield to changes in the rainfall spatial structure, due to the intensification of peak rainfall intensity or due to different rainfall spatial correlation structures is estimated to be in the order of 10%.

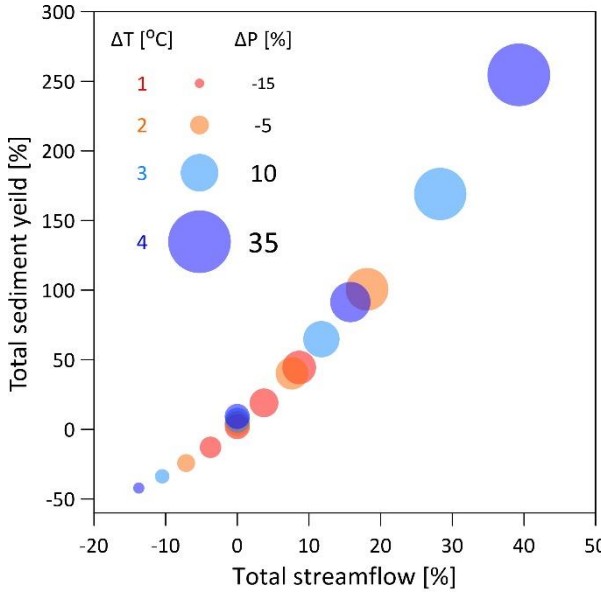

**Figure 9. Changes in the total sediment yield (y-axis) as a function of the changes in total streamflow (x-axis) in relation to the changes in total rainfall amounts (symbol size, ΔP). The relevant temperature scenario (ΔT) is expressed by the color of the symbols**

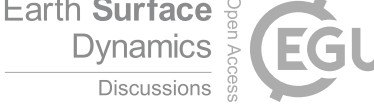

**(1°C – red, 2°C – orange, 3°C – blue and 4°C – purple). The reference to compute the changes is the total streamflow and sediment transport of the base scenario (at ΔT=0).**

The spatial sensitivity of the hydro-morphological response was last examined by means of two spatial indices: (i) the maximal

inundated area for each realization, defined as the total area with water levels higher than 1 cm, which represents the area of

overland flow on hillslopes and in channels; and (ii) the total area of erosion or deposition (active area from hereafter), defined

by comparing the elevation map at the end of the realization to the pre-storm elevation map and summing the grid cells where

net erosion or deposition occurred (i.e. grid cells with elevation difference greater than 1 cm; see example in Fig. S3). Both

the inundated area and active area were found to react to the changes in mean areal rainfall intensity. An increase in areal

rainfall amounts and in the area of heavy rainfall resulted in an increase in both the inundated and active areas (IMF%=3%

and 7%, Fig. 10), while a decrease in areal rainfall amounts and in the area of heavy rainfall resulted in a decrease in both

(IMF%=-3%, Fig. 10). The inundated area and active area has a different sensitivity to the intensification of peak rainfall

intensity for the case when the mean areal rainfall remain unchanged (case 2, IMF%=0%). In this case the inundated area was

found to slightly decrease from 54.5 km$^2$ for ΔT=0°C to 54.3 km$^2$ for ΔT=4°C (Fig. 10a) – practically remaining unchanged

(less than 1% change), while the active area was found to increase from 67.6 km$^2$ for ΔT=0°C to 71 km$^2$ (+5%) for ΔT=4°C

(Fig. 10b).

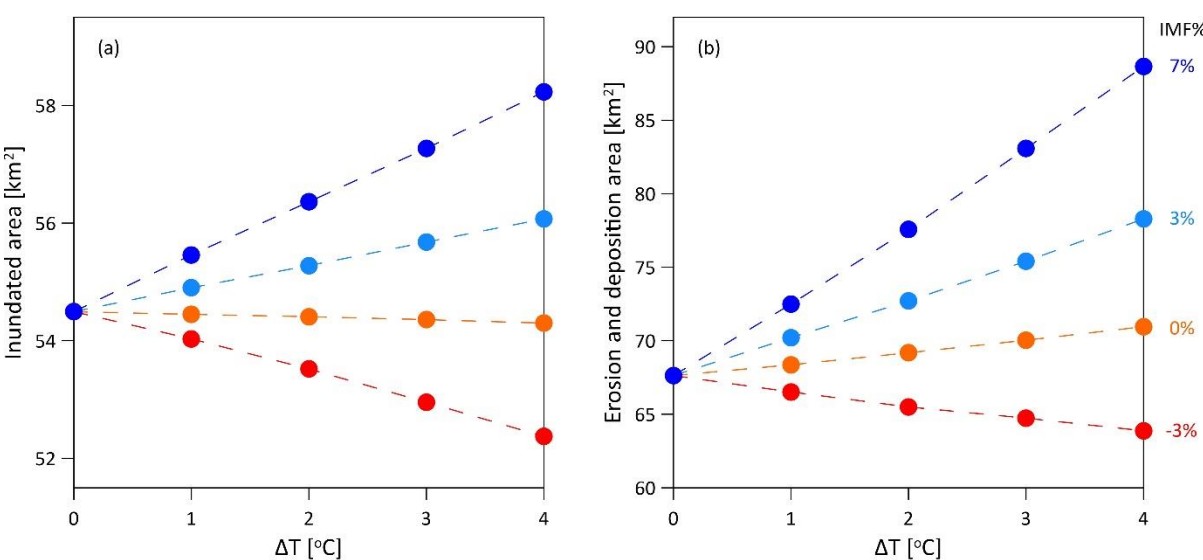

**Figure 10. The sensitivity of the inundated area (a) and the active erosion plus deposition area (b) to changes in the rainfall spatial**

**structure with temperature. Peak rainfall intensity increases by 7% °C$^{-1}$ for all points. Different colors represent mean areal rainfall intensity decrease of -3% °C$^{-1}$ (red), not changing (orange), and increases of 3% °C$^{-1}$ (blue) and 7% °C$^{-1}$ (purple). Reference (base scenario) for the sensitivity is the point at ΔT=0.**





## 4.2 Convective rainfall type

The hydro-morphological response originated by convective rainfall was compared to the response obtained with the stratiform rainfall experiment. Similar trends for the hydro-morphological response were found for the two rainfall types, however, magnitudes were different for some of the examined components. Differences in the hydro-morphological response for the

$\Delta T=2°C$ scenario between the two rainfall types were compared for cases 1 and 3 (Table 4). The hydro-morphological response for the scenario in which both the peak rainfall intensity and mean areal rainfall are intensifying (case 3) is similar for the two rainfall types (Table 4). For the case where the peak rainfall intensified while the mean areal rainfall intensity weakened (case 1), a smaller decrease (between the two rainfall types) in the peak streamflow, peak sediment yield, total sediment yield and in the active area were found for the simulations with the convective rainfall type (Table 4). On the other hand, the decrease

in the inundated area was found to be larger for the convective rainfall type (Table 4).

**Table 4. The hydro-morphological response produced by stratiform and convective rainfall types for the $\Delta T=2°C$ scenario in comparison to the reference scenario ($\Delta T=0°C$).**

|  | Stratiform rainfall | | | Convective rainfall | | |
|---|---|---|---|---|---|---|
|  | Reference | Case 1 | Case 3 | Reference | Case 1 | Case 3 |
| Peak streamflow | 521 m$^3$ s$^{-1}$ | -9.6% | 12.1% | 583 m$^3$ s$^{-1}$ | -5.4% | 13.9% |
| Peak sediment yield | 5.7 m$^3$ s$^{-1}$ | -17.8% | 38.9% | 11.2 m$^3$ s$^{-1}$ | 1.7% | 32% |
| Total streamflow | 104,035 m$^3$ | -7% | 8% | 103,924 m$^3$ | -8% | 7% |
| Total sediment yield | 324 m$^3$ | -24% | 40% | 481 m$^3$ | -8% | 37.3% |
| Inundated area | 54.5 km$^2$ | -1.8% | 1.4% | 52.9 km$^2$ | -5.7% | 0.7% |
| Area of erosion and deposition | 67.6 km$^2$ | -3.2% | 7.5% | 82.1 km$^2$ | -1% | 6.8% |

Differences in the hydro-morphological response were also found for case 2 (no change in areal rainfall, not shown). While for the stratifrom rainfall the peak streamflow remains unchanged and a small increase in the peak sediment yield was reported, for the convective rainfall both peaks considerably increased. The streamflow volume remains unchanged for both rainfall types, while the enhancement of the total sediment yield was found to be more pronounced for the convective rainfall type. In agreement with the stratiform rainfall, the streamflow volume and total sediment yield emerging from the convective rainfall

follow a power law relation (not shown). However, the power function fitted to the stratiform rainfall was less steeper the power function fitted to the convective rainfall, which implies that the 'geomorphic multiplier', besides being depended on the catchment characteristics and sediment supply, depends to a certain extent on rainfall structure.

## 4.3 Summary of the CC experiment

A qualitative summary of the sensitivity of the hydro-morphologic response to the two rainfall types is presented in Table 5.





The response of the hydrological component is ranging from no response (stratiform rainfall) to low positive response (convective rainfall) for the case where intensification of the peak rainfall intensity at the grid scale is the only change considered (case 2), with a low (stratiform) to medium (convective) positive response for the geomorphological components. The response is markedly positive for an increase in the mean areal rainfall (case 3) for both rainfall types, with a medium

enhancement of streamflow volume and peaks and high enhancement of sediment peak and total yield. The response for a decrease in the mean areal rainfall (case 1) is toward a decrease of the analysed variables, but differences can be observed for the two rainfall types. For stratiform rainfall, both the hydrological and geomorphological variables showed a medium negative response, while for the convective rainfall, a medium negative response was detected for the hydrological variables and a lower negative response was found for the geomorphological ones. Results demonstrated that the hydro-morphological response is

sensitive to the rainfall spatial structure as conditioned on the rainfall type, even though rainfall volume remains the most important factor.

**Table 5. A qualitative summary of the hydro-morphological response to the three main studied cases (Fig. 1) for the rainfall spatial structure corresponding to stratiform and convective types. For all three cases, the peak rainfall intensity at the grid scale intensifies.**
**Case (1) total rainfall amount decreases, area of heavy rainfall decreases; (2) total rainfall amount remains unchanged and area of heavy rainfall slightly decreases; and (3) total rainfall amount increases, area of heavy rainfall increases. The response varies between a strong negative response (- - -), no change (o) and a strong positive response (+ + +).**

|  | Stratiform rainfall | | | Convective rainfall | | |
|---|---|---|---|---|---|---|
|  | Case 1 | Case 2 | Case 3 | Case 1 | Case 2 | Case 3 |
| Peak streamflow | - - | o | + + | - | + | + + |
| Peak sediment yield | - - | + | + + + | o | + + | + + + |
| Total streamflow | - - | o | + + | - - | o | + + |
| Total sediment yield | - - | + | + + + | - | + + | + + + |
| Inundated area | - | o | + | - - | o | + |
| Area of erosion and deposition | - - | + | + + + | - | + | + + + |

### 4.3 2CC experiment

An enhancement in the peak streamflow and peak sediment yield were found when increasing the peak rainfall intensity while preserving the mean areal rainfall. For example, for $\Delta T=1°C$, the peak streamflow moderately increased from 583 m$^3$ s$^{-1}$ (reference) to 593 m$^3$ s$^{-1}$ (CC) and 604 m$^3$ s$^{-1}$ (2CC), while the peak sediment yield increased from 11.2 m$^3$ s$^{-1}$ (reference) to 12 m$^3$ s$^{-1}$ (CC) and 12.8 m$^3$ s$^{-1}$ (2CC). In addition, a small decrease in the streamflow volume and a considerable increase in the total sediment yield were observed, respectively reducing from 103,924 m$^3$ (reference) to 103,326 m$^3$ (2CC) and increasing

from 481 m$^3$ (reference) to 535 m$^3$ (2CC). Small decrease was found for the inundated area, from 52.9 km$^2$ (reference) to 51.9 km$^2$ (2CC), while a small increase was observed for the active area, from 82.1 km$^2$ (reference) to 84.5 km$^2$ (2CC).



## 5 Discussion

### 5.1 Physical interpretation of the hydro-morphological responses

As expected, intensification of the peak and mean areal rainfall lead to a higher streamflow peak because the total amount of rainfall considerably increases (Table 4, case 3). However, intensification of the peak rainfall at the grid scale alone, without

changes to the total rainfall amounts (case 2), does not necessarily enhance the peak streamflow as the intensification of the peak rainfall is associated with a decrease in the area of the heavy rainfall over the rainfall field to preserve the rainfall volume. Stratiform rainfall fields are characterized by relatively high spatial correlation in space that is not changing dramatically between the reference and climate scenarios (e.g. Fig. 4). That means that the local intensification of heavy rainfall and decrease of the area of heavy rainfall over the rainfall fields results in minor changes to the volume of rainfall that is reaching the

catchment at a given time, thus the peak streamflow remains largely unchanged (e.g. Bell and Moore, 2000; Gupta et al., 1996; Kalinga and Gan, 2006). However, for the convective rainfall, the change in rainfall spatial structure is expressed by generating high-intensity convective features (Haerter, 2019; Haerter et al., 2017) that are contributing higher volumes of rainfall to specific parts of the catchment in shorter durations, resulting in an enhancement of runoff and peak streamflow (e.g. Morin et al., 2006; Peleg et al., 2015; Yakir and Morin, 2011). When the mean areal rainfall is decreasing but the peak rainfall is

intensifying (Table 4, case 1), for both stratiform and convective rainfall we observed a decrease in the peak streamflow due to the lower rainfall volumes, but such a decrease is much less pronounced for convective rainfall.

Unsurprisingly, the changes in streamflow volume are following the changes in the total rainfall volume, which are expressed by changes in mean areal rainfall (Table 4, cases 1 and 3). However, the streamflow volume is not sensitive to the intensification of the peak rainfall by itself (case 2), as negligible (less than 1%) changes in streamflow volume were produced.

Similarly, the inundated area was found sensitive mainly to the changes in the mean areal rainfall and therefore rainfall volume. A larger decrease in the inundated area is observed for the convective rainfall in comparison to the stratiform rainfall (Table 4, case 1). This is related to the fact that intense convective rainfall features vary significantly in space and sometime are not covering the entire catchment, in such a case not all tributaries are affected by the heavy rainfall (Goodrich et al., 1995; Lopes, 1996), and the total inundated area tends to decrease.

The geomorphological variables, peak and total sediment yield and area of erosion or deposition, were found to be more sensitive to changes in rainfall structure itself in comparison to the hydrological variables (Table 4). This is related to the geomorphic multiplier, the capacity of sediment transport to respond disproportionally to a change in the hydrological regime, and it is the outcome of processes, which are triggered only when certain thresholds are passed. The intensification of the peak rainfall intensity at the grid scale alone (case 2) explains part of the response, as higher rainfall rates are expected to enhance

soil erosion (Nearing et al., 2004, 2005; Shen et al., 2016) and trigger more landslides and debris flows (Guzzetti et al., 2008; Iverson, 2000; Leonarduzzi et al., 2017). Although not explicitly exploring the geomorphological response at the storm scale, Coulthard et al. (2012b), Coulthard and Skinner (2016) and Deal et al. (2017) discussed already the role of local extreme rainfall intensity in enhancing erosion and accelerating landscape evolution. Adding to their conclusions, it is clear that the




geomorphological response is sensitive to the rainfall type as intense convective rainfall features enhance the geomorphological response (Table 4, case 2).

## 5.2 Implications for climate change impact studies

Results imply that the hydro-morphological response of medium size catchments is influenced by changes in the rainfall spatial
structure at small-scale, and due to above-mentioned threshold effects, the geomorphological variables are more sensitive to changes in rainfall structure than the hydrological variables. Four representative cases of changes in rainfall spatial structure with increasing temperature were explored in order to study the sensitivity of the fluvial system to these changes. Likely, the sensitivity of the hydro-morphological response in reality is larger than presented here, as the changes in rainfall structure are likely to be more complex than schematized in this work. For example, Wasko et al. (2016) showed that the intensification of
the peak rainfall is associated with a reduction in the area of the storm and with an intensification of the mean areal rainfall, but with a lower rate than the intensification in the rainfall peak.

Representing spatial rainfall structures requires high-resolution simulations. In hydrology, using gridded rainfall data for climate change impact studies is becoming common practice. In recent years, there was an increase in the availability of distributed hydrological models (Fatichi et al., 2016; Paniconi and Putti 2015) and they have been used in the context of climate
impacts (e.g. Dams et al., 2015; Fatichi et al., 2015; Perra et al., 2018). However, this is much less the case for geomorphological impacts (e.g. Francipane et al., 2015; Pandey et al., 2016; Ramsankaran et al., 2013; Zi et al., 2016), where the added value of using high-resolution gridded rainfall data for climate change impact studies is still not widely evaluated (Li and Fang, 2016) and most LEMs do not receive distributed rainfall as input (Coulthard and Skinner, 2016; Tucker and Hancock, 2010). The risk of over-predicting the geomorphological response is significantly increased when using uniform
rainfall instead of distributed rainfall. This was discussed by Coulthard and Skinner (2016), and here we demonstrate plausible consequences of global warming on the spatial rainfall structure and its potential impacts. Specifically, case 4 represents the situation where both peak rainfall intensity and mean areal intensity are assumed to intensify at the same rate, meaning that the rainfall amounts are increasing while the rainfall spatial structure remains largely unchanged. In the context of a climate change study, using case 4 would be equivalent to using the observed rainfall and increase the rainfall amounts evenly over
the catchment in response to climate change. The peaks of streamflow and sediment yield (Fig. 8), streamflow volume and total sediment yield (Fig. 9) and inundated area and active area (Fig. 10) - all these quantities - will be over-predicted for such a case in comparison to other scenarios, where the areal and peak rainfall intensity, and thus the rainfall spatial structure, are changing at different rates. This also emphasizes the importance of either using high-resolution CPMs to simulate changes in the rainfall for the future climate or in downscaling the rainfall simulations from RCMs. If downscaling is based on simple
relationships between rainfall and temperature that are measured at the point scale (i.e. using ground stations), results must be analysed with care, as often these relationships are extrapolated to a much larger (catchment) scale (Dahm et al., 2019; Fadhel et al., 2018; Lenderink and Attema, 2015). A viable alternative presented here. It is represented by space-time rainfall generators, which allow downscaling the rainfall to the required space-time resolution while maintaining spatial characteristics





of rainfall fields based on current observations at a point and weather radar scales and factors of change derived from climate models (e.g. Peleg et al., 2019). We note that information on the relation between the rainfall spatial structure and temperatures were observed so far only in few locations worldwide (Lochbihler et al., 2017; Peleg et al., 2018b; Wasko et al., 2016), and they remain unknown for most locations.

## 5.3 Generalization and limitations of the numerical experiment

The presented study offers a quite large evaluation of different spatial structures of rainfall and their effects on key hydro-morphological variables, but it also refers to a single design storm, a single catchment, and only one LEM was used. The design storm represents a realistic storm in terms of rainfall amount, but it is simplified in terms of the temporal evolution, which can affect the fluvial response (Istanbulluoglu and Bras, 2006; Tucker and Bras, 2000); furthermore, the storm is advected with a fixed velocity and direction and the rainfall is assumed to cover the entire catchment – assumptions that can affect the hydrograph peak, volume, and timing (Morin et al., 2006; Singh, 1997; Yakir and Morin, 2011). The LEM was used with a simplified setup that does not include vegetation-erosion interactions and lateral erosion; concurrently, the hydrological module is not representing all of the hydrological processes, such as changes in soil saturation and infiltration capacity, which can be important for precise estimation of the hydro-morphological response (Coulthard et al., 2013; Paschalis et al., 2014; Tucker and Hancock, 2010). The TOPMODEL formulation of runoff production, controlled exclusively by the saturated area that is proportional to the rainfall volume, scales the streamflow volume almost linearly with the rainfall volume. Assumptions that are more realistic may provide a more pronounced non-linear response of the hydrology of the catchment and therefore emphasize the role of rainfall spatial variability. However, the LEM validation proved that the model simulates streamflow and sediment yield satisfactory for the examined 15-days period, which generate confidence on the overall predictions.

Even though we use a single catchment, the Kleine Emme catchment can be considered representative of an average Alpine catchment in terms of area, elevation, land-use, and stream morphology. Therefore, the sensitivities of the hydro-morphological response to different spatial representations of heavy rainfall found in this study are likely indicative of the behaviour of other medium-size mountainous catchments. Extrapolations for other climatic regions, rainfall regimes, catchments of with a different topography and geomorphology are instead highly speculative.

In this study, we considered sediment yield at the outlet of the catchment and the total area of erosion and deposition within the catchment as a representative function of the geomorphic change to rainfall structure. Further internal geomorphological changes (see Skinner et al., 2018) can be explored and are subject of future work.

Finally, while we studied the small-scale spatial structure of rainfall, additional research is needed in order to understand the sensitivity of the catchment to the rainfall temporal structure at fine-scale and its relative importance when compared to spatial structure in shaping the landscape over long (millennial) periods.

**6 Conclusions**

A numerical experiment was conducted to examine the sensitivity of the hydro-morphological response to changes in rainfall spatial structure, as plausibly modified by increasing temperatures for stratiform and convective rainfall types. Results demonstrated that hydrological and sediment related variables are sensitive to changes in the rainfall spatial structure, with a
much higher sensitivity for the morphological components in comparison to the hydrological components, due to responses activated only when certain thresholds (rainfall intensity for erosion, discharge for sediment transport) are exceeded (the geomorphic multiplier), while hydrological processes in this catchment are simulated to scale almost linearly with rainfall amount, which can be partially due to the simplified runoff generation scheme that is used. Regardless of uncertainty, predicting changes in hydro-morphological response require plausible scenarios of how both the peak rainfall intensity at
small-spatial scale and the mean areal rainfall of heavy storms are going to change, as was done here. Neglecting changes in the rainfall spatial structure and assuming that the mean areal and peak rainfall intensity will follow a similar scaling with increasing temperatures may be misleading and lead to over-prediction of the hydro-morphological response. This effect is more pronounced for convective rainfall that produce more localised runoff and erosion potential.

*Data and models availability.* All the data used in this study is freely available and can be collected from the relevant agencies that are mentioned in Section 3.1 and in the acknowledgments below. The MATLAB source code of the rainfall generator model (or the full version of the AWE-GEN-2d model) is available upon request from NP. Any element of this code is free to use, modify, copy or distribute provided it is for academic use and source code developers are properly acknowledged and cited. The CAESAR-Lisflood model is freely available for academic use; Windows-based compiled version of the model and
the source code (C sharp) can be downloaded from https://sourceforge.net/projects/caesar-lisflood/.

*Author contribution.* NP carried out the numerical experiment, produced the figures, and wrote the manuscript. CS assisted in setting the CAESAR-Lisflood model and guidance in the geomorphic analysis. NP, SF, and PM designed the project. All authors contributed to the manuscript edits.

*Competing interests.* The authors declare that they have no conflict of interest.

*Financial support.* NP is funded by the Swiss Competence Center for Energy Research - Supply of Electricity (http://www.sccer-soe.ch).

*Acknowledgments.* We thank MeteoSwiss, the Swiss Federal Office of Meteorology and Climatology, for supplying climate data from ground stations and gridded products, the Swiss Federal Office for the Environment for supplying the streamflow data, the Swiss Federal Statistical Office for supplying the land use map and the Swiss Federal Office of Topography for



supplying the digital elevation model and the bedrock map. NP thanks Giulia Battista for many fruitful discussions during the preparation of this paper.

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
