# Peer review of "Temperature effects on the spatial structure of heavy rainfall modify catchment hydro-morphological response"

_Earth Surface Dynamics, 2019_

## Referee Comment (RC1) · Anonymous Referee #1 · 8 Oct 2019

As far as I am aware this is the first manuscript examining the effects of changes to spatial rainfall patterns under climate change. I believe this is a very important contribution because, as interesting rainfall changes are, it is the on-ground impacts that are really of key importance. Some small comments below (suggestions are at the authors discretion).

Minor comments:

Section 4.2: I got a bit confused here because I thought case 3 could be either where mean rainfall increases by 3% or 7%. I only realized what case 4 was in Section 5.2. Can "case 4" be added to Figure1 and the cases (1-4) be labelled in Table 1?

[Figure]

I don't entirely agree with there just being two options for spatial rainfall downscaling (i.e. CPM or rain-temperature sensitivities). Some authors use stochastic methods (Bordoy and Burlando, 2014) others use RCMs directly with varying convective parametrizations which will affect the results (Li et al., 2018).

Bordoy, R., Burlando, P., 2014. Stochastic downscaling of climate model precipitation outputs in orographically complex regions: 2. Downscaling methodology. Water Resour. Res. 50, 562–579. https://doi.org/10.1002/wrcr.20443

Li, J., Wasko, C., Johnson, F., Evans, J.P., Sharma, A., 2018. Can Regional Climate Modeling Capture the Observed Changes in Spatial Organization of Extreme Storms at Higher Temperatures? Geophys. Res. Lett. 45, 4475–4484. https://doi.org/10.1029/2018GL077716

Line by line:

**Title: add "spatial structure" somewhere?**

**Page 1, Line 11: "small-scale resolution . . ."**

**Page 3, Line 16: "The second option" -> "An alternative"**

**Page 4, Line 6: "on the hydro. . ."**

**Page 9, Line 13: "is simulated" -> "can be simulated" (because later you choose one method)**

**Figure 5 caption: "km" -> "m"**

**Page 12, Line 15: Could you add the domain to Figure 5?**

**Page 13, Line 5: "corresponds" -> "corresponding"**

**Page 15, Line14-18: Could this be moved up to follow the first sentence in this section? Just fits better with the point made in the first sentence.**

**Figure 8 caption: "black" -> "grey"?**

**Page 16., Line 18: 10% of what?**

**Figure 9: could add zero, zero guide lines**

**Page 18, Line 17: "considerably increased". Feels like this result is interesting and could easily be included in Table 4?**

**Page 19, Line 9: "demonstrate" -> "demonstrated"**

**Page 19, Line 19: Section "4.4"**

**Page 21, Line 4: "by small-scale changes in the rainfall spatial structure. . ."**

**Page 21, Line 33: "A viable alternative presented here is . . ."**

**Page 22, Line 7: You use 10 stochastic replicates – could mention this here.**

**Page 23, Line 10: Add the words "under climate change" before "as was done here"?**

**Figure S3: can you add to the legend if this is an example of a stratiform or convective storm?**

---

## Referee Comment (RC2) · Anonymous Referee #2 · 11 Oct 2019

I have been invited to review "Temperature effects on heavy rainfall modify catchment hydro- morphological response". My expertise is in statistics, landscape evolution modeling and sediment transport models. Therefore I feel well qualified to evaluate the relevance, novelty, and quality of this work, with caveat that I have little expertise in sophisticated stochastic rainfall generators, and cannot evaluate those sections of this work.

The authors explore the effects of increasing rainfall intensity on sediment transport processes in a small alpine catchment through use of several coupled simulations. They correctly point out that most existing landscape evolution models (LEMs - which

model sediment transport) implement rainfall forcing in only the most basic way, and in order to address the question they have posed, it is necessary to have a landscape evolution that incorporates spatially distributed rainfall and hydrology, no easy feat. A large part of the novelty in this work stems from their use of cutting edge LEMs that are capable of simulating the geomorphic response to spatial and temporal changes in the rainfall forcing in a realistic way. Their integration of a sophisticated 2-D stochastic rainfall generator is also novel, and allows them to tackle the question of increasing rainfall intensity in a credible way. Though the timescales in question are small, much less than the timescales that geomorphologists usually think about, the relevance of their study is high due to the robust predictions of increasing rainfall intensity with increasing atmospheric temperatures, and concerns about how that might impact sediment transport processes with implications for geohazards.

In addition to the work being novel and relevant, I find the methodological choices to be reasonable, and I find no noteworthy faults with the approaches taken here (except for the details of the rainfall generator, which I do not have the expertise to evaluate). The work is timely and very well referenced, the authors are clearly aware of the state of the art in the relevant fields. The one significant criticism I have is that I think the authors could be a bit more clear in the abstract and conclusions about the result that peak rainfall intensity has little effect on the geomorphic response, especially in comparison to changes in the mean areal rainfall rates. I have highlighted those sections below along with some other minor suggestions. I think these suggestions would improve the manuscript, however I don't think that any are absolutely required, and it could be published as is.

Details: Abstract - Lines 16-18: " The results highlight that the response of the streamflow and sediment yields are highly sensitive to changes in the rainfall structure at the small-scale, in particular to changes in the areal rainfall intensity and in the area of heavy rainfall, which alter the total rainfall volume, and to a lesser extent to changes in the peak rainfall intensity." - This claim feels a little misleading to me. The use of

"rainfall structure" in the first sentence implies that something complicated is going on, when really it is just changes in areal intensity - can that count as rainfall structure? Similarly, "to a lesser extent" is not clear about the fact that the areal rainfall intensity is by far the dominant variable compared to peak intensity. You later make the point that simpler models that just increase the rainfall pattern as is without taking into account the seperate effects of mean areal intensity and peak intensity related through storm structure will over predict erosion. It seems that you're showing the large scale volumes are the first order effect, so actually simple models would get it right, as long as they don't confuse peak intensity with areal mean intensity. It seems to me it would be more useful to point out the importance of changing rainfall intensity at the correct scale. Abstract - Lines 18-19: "The hydro-morphological response is enhanced (reduced) when the local peak rainfall intensified and the area of heavy rainfall increased (decreased)" - It seems something is missing. Writing out the two versions of the sentence: The hydro-morphological response is enhanced when the local peak rainfall intensified and the area of heavy rainfall increased. The hydro-morphological response is (reduced) when the local peak rainfall intensified and the area of heavy rainfall (decreased). Local peak rainfall is missing it's paranthetical partner. Abstract - Line 9: "and how those impactS" - change to something like "and subsequent impacts on" Page 9 - Line 8: "moving sediments that are stored" -> "moving sediment that is stored" Page 9 - Line 24: "variables than rainfall" -> "variables besides rainfall" Page 11 - line 20: Drop paran- theses on citation Page 16 - line 13: I was hoping you would share the exponent - could you add it - the convective rainfall one too? Page 18 - line 2: "The hydro-morphological response originated" -> "The hydro-morphological response driven/caused" Page 18 - lines 18-22: As I read it, these lines explain that convective rainfall drives a more pro- nounced enhancement of the total sediment yield, yet at the same time, the sensitivity of change in sediment yield to change in flow is weaker. This is a bit confusing. On the surface these seem like opposing statements. Is this because, while the exponent is smaller, the change in streamflow is so much greater than for stratiform rainfall that the overall increase in sediment yield is still greater? Is it possible to give a bit more explanation here? Page 21 - lines 7 - 9: "Likely, the sensitivity of the hydro-morphological response in reality is larger than presented here, as the changes in rainfall structure are likely to be more complex than schematized in this work" - this is a confusing sentence because you argued in the abstract, and again directly after this statement that the real danger in simple models is in overprediction. But now you say that a limitation of the sophisticated model is that it underpredicts by an unquantified amount... Page 21 - line 32: "A viable alternative presented here" -> "A viable alternative is presented here" Page 22 - lines 23-24: "catchments of with a different topography" -> "catchments with a different topography" Page 22 - line 27: "are subject of future work." -> "are the/will be the subject of future work." Page 23 - line 4: "sediment related" -> "sediment transport/geomorphological/morphological related" Page 23 - line 5: "morphological" match to previous use of word in same sentence (previous suggestion) for clarity Page 23 - line 6: "erosion" - it would be better to use a more specific term, erosion happens by many processes and also in rivers. Perhaps sheet-flow driven erosion, or mass movements or runoff driven erosion

---

## Author Comment (AC1) · 1 Nov 2019

**Reviewer #1**

As far as I am aware this is the first manuscript examining the effects of changes to spatial rainfall patterns under climate change. I believe this is a very important contribution because, as interesting rainfall changes are, it is the on-ground impacts that are really of key importance. Some small comments below (suggestions are at the authors discretion).

We thank the reviewer for their time and effort and for their appreciation of our work.

Minor comments:

Section 4.2: I got a bit confused here because I thought case 3 could be either where mean rainfall increases by 3% or 7%. I only realized what case 4 was in Section 5.2. Can "case 4" be added to Figure1 and the cases (1-4) be labelled in Table 1?

It is a good suggestion - the labels for cases 1 to 4 will be added to Table 1. Adding the line of case 4 in Figure 1 will make the figure less clear due to the overlaps of the new line with some of the original lines.

I don't entirely agree with there just being two options for spatial rainfall downscaling (i.e. CPM or rain-temperature sensitivities). Some authors use stochastic methods (Bordoy and Burlando, 2014) others use RCMs directly with varying convective parametrizations which will affect the results (Li et al., 2018).

The study by Bordoy and Burlando suggests a methodology to downscale rainfall at a sub-hourly scale and for multiple locations, without considering the storm internal structure of rainfall at small scales as discussed in this paper. The work by Li et al. is interesting and relevant as it discusses the potential use of RCMs to explicitly represent some of the rainfall properties in space, although at coarser scales than discussed in this paper. We will add a reference to the study by Li et al.

Line by line:

**Title: add "spatial structure" somewhere?**

We will modify the title following the suggestion of the reviewer. The new titled will be: "Temperature effects on the spatial structure of heavy rainfall modify catchment hydro-morphological response".

**Page 1, Line 11: "small-scale resolution . . ."**
**Page 3, Line 16: "The second option" -> "An alternative"**
**Page 4, Line 6: "on the hydro. . ."**
**Page 9, Line 13: "is simulated" -> "can be simulated" (because later you choose one method)**
**Figure 5 caption: "km" -> "m"**
**Page 13, Line 5: "corresponds" -> "corresponding"**
**Figure 8 caption: "black" -> "grey"?**
**Page 19, Line 9: "demonstrate" -> "demonstrated"**
**Page 19, Line 19: Section "4.4"**
**Page 21, Line 4: "by small-scale changes in the rainfall spatial structure. . ."**
**Page 21, Line 33: "A viable alternative presented here is . . ."**
**Page 23, Line 10: Add the words "under climate change" before "as was done here"?**

We thank the reviewer for pointing on typos and suggesting text edits. All issues will be resolved in the revised manuscript.

**Page 12, Line 15: Could you add the domain to Figure 5?**

The simulated domain will be added to Figure 5.

\# Page 15, Line14-18: Could this be moved up to follow the first sentence in this section? Just fits better with the point made in the first sentence.

We prefer to retain the structure of this section as it is, namely discussing first the example of the impacts of changing the mean areal intensity on the peak streamflow and sediment yield and at the end discussing the impacts emerging from changes in the rainfall peak intensity.

\# Page 16., Line 18: 10% of what?

From the reference total sediment yield. The information will be added to the text.

\# Figure 9: could add zero, zero guide lines

Guidelines will be added to the figure, as suggested.

\# Page 18, Line 17: "considerably increased". Feels like this result is interesting and could easily be included in Table 4?

We agree. We will add the results in Table 4, as suggested.

\# Page 22, Line 7: You use 10 stochastic replicates – could mention this here.

Thank you for this suggestion, the text will be revised to mention that 10 stochastic realizations were used.

\# Figure S3: can you add to the legend if this is an example of a stratiform or convective storm?

The example is of a stratiform storm. This will be added to the figure caption.

Finally, we would like again to express our deepest thanks to the Associate Editor and to the two reviewers who have helped us to significantly improve the paper.

Sincerely,
Nadav Peleg, Chris Skinner, Simone Fatichi and Peter Molnar

---

## Author Response (AR1)

November 5th, 2019

Prof. Dr. Jean Braun
Associate Editor
Earth Surface Dynamics

RE: Paper ESurf-2019-44

Dear Prof. Braun,

Please find enclosed the revised manuscript "Temperature effects on heavy rainfall modify catchment hydro-morphological" by Nadav Peleg, Chris Skinner, Simone Fatichi and Peter Molnar. The manuscript has been revised according to the comments of the two reviewers. We would like to thank the Editor, Associate Editor and the two anonymous reviewers for their efforts and constructive comments. Detailed answers to the reviewers' comments are provided below.

**Reviewer #1**
As far as I am aware this is the first manuscript examining the effects of changes to spatial rainfall patterns under climate change. I believe this is a very important contribution because, as interesting rainfall changes are, it is the on-ground impacts that are really of key importance. Some small comments below (suggestions are at the authors discretion).
We thank the reviewer for their time and effort and for their appreciation of our work.

Minor comments:
Section 4.2: I got a bit confused here because I thought case 3 could be either where mean rainfall increases by 3% or 7%. I only realized what case 4 was in Section 5.2. Can "case 4" be added to Figure1 and the cases (1-4) be labelled in Table 1?
It is a good suggestion - the labels for cases 1 to 4 were added to Table 1. Adding the line of case 4 in Figure 1 made the figure unclear due to the overlaps of the new line with some of the original lines.

I don't entirely agree with there just being two options for spatial rainfall downscaling (i.e. CPM or rain-temperature sensitivities). Some authors use stochastic methods (Bordoy and Burlando, 2014) others use RCMs directly with varying convective parametrizations which will affect the results (Li et al., 2018).
The study by Bordoy and Burlando suggests a methodology to downscale rainfall at sub-hourly scale and for multiple locations, without considering the storm internal structure of rainfall at small scales as discussed in this paper. The work by Li et al. is interesting and relevant as it discusses the potential use of RCMs to explicitly represent some of the rainfall properties in space, although at coarser scales than discussed in this paper. The following text was added in the introduction part: "RCMs simulate rainfall fields at a spatial resolution not far from what is needed in local impact studies … but they do not resolve convection processes explicitly. Nevertheless, some of the rainfall spatial properties can be properly represented by RCMs if appropriate convective parametrizations are used, as recently discussed by Li et al. (2018) …".

Line by line:
**Title: add "spatial structure" somewhere?**
We edited the title following the suggestion of the reviewer. The new titled is: "Temperature effects on the spatial structure of heavy rainfall modify catchment hydro-morphological response".

**Page 1, Line 11: "small-scale resolution . . ."**
The word "resolution" was added to the text.

**Page 3, Line 16: "The second option" -> "An alternative"**
The text was modified as suggested.

**Page 4, Line 6: "on the hydro. . ."**
The word "the" was added to the text.

**Page 9, Line 13: "is simulated" -> "can be simulated" (because later you choose one method)**
Corrected, as suggested.

**Figure 5 caption: "km" -> "m"**
Corrected.

**Page 12, Line 15: Could you add the domain to Figure 5?**
The simulated domain was added to Figure 5.

**Page 13, Line 5: "corresponds" -> "corresponding"**
Corrected.

**Page 15, Line14-18: Could this be moved up to follow the first sentence in this section? Just fits better with the point made in the first sentence.**
We prefer to retain the structure of this section as it is, namely discussing first the example of the impacts of changing the mean areal intensity on the peak streamflow and sediment yield and at the end discussing the impacts emerging from changes in the rainfall peak intensity (i.e. lines 14 to 18).

**Figure 8 caption: "black" -> "grey"?**
Grey indeed, the text was corrected.

**Page 16., Line 18: 10% of what?**
From the reference total sediment yield. The information was added to the text.

**Figure 9: could add zero, zero guide lines**
Guide lines were added to the figure, as suggested.

**Page 18, Line 17: "considerably increased". Feels like this result is interesting and could easily be included in Table 4?**
Results are now included in Table 4, as suggested.

**Page 19, Line 9: "demonstrate" -> "demonstrated"**
The word "demonstrated" is used in the sentence.

**Page 19, Line 19: Section "4.4"**
Corrected.

**Page 21, Line 4: "by small-scale changes in the rainfall spatial structure. . ."**
We prefer to keep the sentence as it is.

**Page 21, Line 33: "A viable alternative presented here is . . ."**
The sentence was modified as follows: "A viable alternative is presented here".

**Page 22, Line 7: You use 10 stochastic replicates – could mention this here.**
Thank you for this suggestion, the text was revised to mention that 10 stochastic realizations were used.

**Page 23, Line 10: Add the words "under climate change" before "as was done here"?**
The original sentence was changed.

**Figure S3: can you add to the legend if this is an example of a stratiform or convective storm?**
The example is of a stratiform storm. This was added to the figure caption.

**Reviewer #2**
I have been invited to review "Temperature effects on heavy rainfall modify catchment hydro-morphological response". My expertise is in statistics, landscape evolution modeling and sediment transport models. Therefore I feel well qualified to evaluate the relevance, novelty, and quality of this work, with caveat that I have little expertise in sophisticated stochastic rainfall generators, and cannot evaluate those sections of this work.

The authors explore the effects of increasing rainfall intensity on sediment transport processes in a small alpine catchment through use of several coupled simulations. They correctly point out that most existing landscape evolution models (LEMs – which model sediment transport) implement rainfall forcing in only the most basic way, and in order to address the question they have posed, it is necessary to have a landscape evolution that incorporates spatially distributed rainfall and hydrology, no easy feat. A large part of the novelty in this work stems from their use of cutting edge LEMs that are capable of simulating the geomorphic response to spatial and temporal changes in the rainfall forcing in a realistic way. Their integration of a sophisticated 2-D stochastic rainfall generator is also novel, and allows them to tackle the question of increasing rainfall intensity in a credible way. Though the timescales in question are small, much less than the timescales that geomorphologists usually think about, the relevance of their study is high due to the robust predictions of increasing rainfall intensity with increasing atmospheric temperatures, and concerns about how that might impact sediment transport processes with implications for geohazards.

In addition to the work being novel and relevant, I find the methodological choices to be reasonable, and I find no noteworthy faults with the approaches taken here (except for the details of the rainfall generator, which I do not have the expertise to evaluate). The work is timely and very well referenced, the authors are clearly aware of the state of the art in the relevant fields. The one significant criticism I have is that I think the authors could be a bit more clear in the abstract and conclusions about the result that peak rainfall intensity has little effect on the geomorphic response, especially in comparison to changes in the mean areal rainfall rates. I have highlighted those sections below along with some other minor suggestions. I think these suggestions would improve the manuscript, however I don't think that any are absolutely required, and it could be published as is.

We thank the reviewer for their time and effort and for their appreciation of our work.

Details:

Abstract - Lines 16-18: " The results highlight that the response of the streamflow and sediment yields are highly sensitive to changes in the rainfall structure at the small-scale, in particular to changes in the areal rainfall intensity and in the area of heavy rainfall, which alter the total rainfall volume, and to a lesser extent to changes in the peak rainfall intensity." - This claim feels a little misleading to me. The use of "rainfall structure" in the first sentence implies that something complicated is going on, when really it is just changes in areal intensity - can that count as rainfall structure? Similarly, "to a lesser extent" is not clear about the fact that the areal rainfall intensity is by far the dominant variable compared to peak intensity. You later make the point that simpler models that just increase the rainfall pattern as is without taking into account the seperate effects of mean areal intensity and peak intensity related through storm structure will over predict erosion. It seems that you're showing the large scale volumes are the first order effect, so actually simple models would get it right, as long as they don't confuse peak intensity with areal mean intensity. It seems to me it would be more useful to point out the importance of changing rainfall intensity at the correct scale. Lines 18-19: "The hydro-morphological response is enhanced (reduced) when the local peak rainfall intensified and the area of heavy rainfall increased (decreased)" - It seems something is missing. Writing out the two versions of the sentence: The hydro-morphological response is enhanced when the local peak rainfall intensified and the area of heavy rainfall increased. The hydro-morphological response is (reduced) when the local peak rainfall intensified and the area of heavy rainfall (decreased). Local peak rainfall is missing it's parenthetical partner.

We agree with the comments made by the reviewer and revised the abstract extensively. Major changes to the text: "… The experiment was conducted over a complex topography, medium-sized (477 km2), Alpine catchment in central Switzerland. It was found that the response of the streamflow and sediment yields are highly sensitive to changes in total rainfall volume and to a lesser extent to changes in local peak rainfall intensities. The results highlight that the morphological components are more sensitive to changes in rainfall spatial structure in comparison to the hydrological components. The hydro-morphological features were found to respond more to convective rainfall than stratiform rainfall because of localized runoff and erosion production …".

Abstract - Line 9: "and how those impacts" - change to something like "and subsequent impacts on"
Changed, as suggested.

Page 9 - Line 8: "moving sediments that are stored" -> "moving sediment that is stored"
Corrected.

Page 9 – Line 24: "variables than rainfall" -> "variables besides rainfall"
Changed, as suggested.

Page 11 - line 20: Drop parentheses on citation
Corrected.

Page 18 - line 2: "The hydro-morphological response originated" -> "The hydro-morphological response driven/caused"
The word "originated" was replaced with "driven".

Page 16 - line 13: I was hoping you would share the exponent – could you add it - the convective rainfall one too?

The functions are: $8.1 \times 10^{-17} Q^{3.7}$ for the stratiform rainfall and $6.5 \times 10^{-13} Q^{3}$ for the convective rainfall, where Q is the total discharge. We added this information in Section 4.2 (see the answer to the next comment), where both stratiform and convective rainfalls are discussed.

Page 18 - lines 18-22: As I read it, these lines explain that convective rainfall drives a more pronounced enhancement of the total sediment yield, yet at the same time, the sensitivity of change in sediment yield to change in flow is weaker. This is a bit confusing. On the surface these seem like opposing statements. Is this because, while the exponent is smaller, the change in streamflow is so much greater than for stratiform rainfall that the overall increase in sediment yield is still greater? Is it possible to give a bit more explanation here?

Convective rainfall does drive a more pronounced enhancement to total sediment yield. For example, an increase from discharge of $100 \times 10^{3}$ m$^3$ to $150 \times 10^{3}$ m$^3$ will results (based on the relations from the previous answer) in an increase in total sediment yield of $1.03 \times 10^{3}$ m$^3$ and $1.14 \times 10^{3}$ m$^3$ for stratiform and convective rainfall, respectively. The text was revised to make this point clearer: "The streamflow volume remains unchanged for both rainfall types, while the enhancement of the total sediment yield was found to be more pronounced for the convective rainfall type. As for the stratiform rainfall, the streamflow volume and total sediment yield simulated with the convective rainfall are related by a power law relation ($S=8.1 \times 10^{-17} Q^{3.7}$ and $S=6.5 \times 10^{-13} Q^{3}$ for stratiform and convective rainfalls, respectively, where S is total sediment yield and Q is total discharge, $R^2$=0.99 for both cases). The difference in the exponential relations between the two rainfall types implies that the 'geomorphic multiplier', besides being depended on the catchment characteristics and sediment supply, depends to a certain extent on rainfall structure as well".

Page 21 - lines 7 - 9: "Likely, the sensitivity of the hydro-morphological response in reality is larger than presented here, as the changes in rainfall structure are likely to be more complex than schematized in this work" - this is a confusing sentence because you argued in the abstract, and again directly after this statement that the real danger in simple models is in over-prediction. But now you say that a limitation of the sophisticated model is that it under-predicts by an unquantified amount...

We didn't mean to imply that weather generator (sophisticated) models under-predict the rainfall spatial patterns, but intended to point out the fact that we examined a number of spatial rainfall scenarios in this study, and didn't cover all options. We rephrased the sentence: "Various representative cases of changes in rainfall spatial structure with increasing temperature were explored in order to study the sensitivity of the fluvial system to these changes. We note that the changes in rainfall structure can be more complex than schematized in this work. For example, Wasko et al. (2016) showed that the intensification of the peak rainfall is associated with a reduction in the area of the storm and with an intensification of the mean areal rainfall, but at a lower rate than the intensification in the rainfall peak".

Page 21 - line 32: "A viable alternative presented here" -> "A viable alternative is presented here"
Corrected.

Page 22 - lines 23-24: "catchments of with a different topography" -> "catchments with a different topography"
The word "of" was deleted.

Page 22 - line 27: "are subject of future work." -> "are the/will be the subject of future work."
Corrected.

Page 23 - line 4: "sediment related" -> "sediment transport/ geomorphological/morphological related"
Page 23 - line 5: "morphological" match to previous use of word in same sentence (previous suggestion) for clarity
The word "sediment" in page 23, line 4 was replaced with "morphological".

Page 23 - line 6: "erosion" - it would be better to use a more specific term, erosion happens by many processes and also in rivers. Perhaps sheet-flow driven erosion, or mass movements or runoff driven erosion
We replaced the term "erosion" with "runoff driven erosion", as suggested.

Finally, we would like again to express our deepest thanks to the Associate Editor and to the two reviewers who have helped us to significantly improve the paper.

Sincerely,
Nadav Peleg, Chris Skinner, Simone Fatichi and Peter Molnar

[revised manuscript text omitted]